# Seroprevalence trends of anti-SARS-CoV-2 antibodies in the adult population of the São Paulo Municipality, Brazil: Results from seven serosurveys from June 2020 to April 2022. The SoroEpi MSP Study

Beatriz Helena Tess [1☯¶]*, Celina Maria Turchi Martelli [2☯¶], Maria Cecília Goi Porto Alves [3¶], Fanny Cortes [4], Regina Tomie Ivata Bernal [5], Wayner Vieira de Souza [2], Expedito José de Albuquerque Luna [1], Laura da Cunha Rodrigues [6], Marcia Cavallari Nunes [7¶], Fernando de Castro Reinach [8¶], Celso Francisco Hernandes Granato [9¶], Edgar Gil Rizzatti [9¶], Maria Carolina Tostes Pintão [9¶]

1 Departamento de Medicina Preventiva, Faculdade de Medicina, Universidade de São Paulo, São Paulo, SP, Brasil, 2 Instituto Aggeu Magalhães, Fiocruz, Recife, PE, Brasil, 3 Instituto de Saúde, Secretaria de Saúde do Estado de São Paulo, São Paulo, SP, Brasil, 4 Programa de Pós-Graduação em Ciências da Saúde, Universidade de Pernambuco, Recife, PE, Brasil, 5 Programa de Pós-Graduação, Escola de Enfermagem, Universidade Federal de Minas Gerais, Belo Horizonte, MG, Brasil, 6 Department of Infectious Disease Epidemiology, London School of Hygiene & Tropical Medicine, London, United Kingdom, 7 Inteligência em Pesquisa e Consultoria (IPEC), São Paulo, SP, Brasil, 8 Departamento de Bioquímica, Instituto de Química, Universidade de São Paulo, São Paulo, SP, Brasil, 9 Divisão de Pesquisa e Desenvolvimento, Grupo Fleury, São Paulo, SP, Brasil

☯ These authors contributed equally to this work.
¶ Members of the SoroEpi MSP Study Research Group.
* beatriz.tess@usp.br

## Abstract

### Background

Sequential population-based household serosurveys of SARS-CoV-2 covering the COVID-19 pre- and post-vaccination periods are scarce in Brazil. This study investigated seropositivity trends in the municipality of São Paulo.

### Methods

We conducted seven cross-sectional surveys of adult population-representative samples between June 2020 and April 2022. The study design included probabilistic sampling, test for SARS-CoV-2 antibodies using the Roche Elecsys anti-nucleocapsid assay, and statistical adjustments for population demographics and non-response. The weighted seroprevalences with 95% confidence intervals (CI) were estimated by sex, age group, race, schooling, and mean income study strata. Time trends in seropositivity were assessed using the Joinpoint model. We compared infection-induced seroprevalences with COVID-19 reported cases in the pre-vaccination period.

**Data Availability Statement:** All relevant data are within the manuscript, all files are available from the SoroEpi-MSP.xlsx database and are acessible at: https://sites.usp.br/epi-di/pagina-1/pagina-1-1/.

**Funding:** The SoroEpi MSP Study was funded by the Instituto Semeia, Todos pela Saúde, The Grupo Fleury and IBOPE/IPEC. The current analysis did not receive specific funding. Instituto Semeia and Todos pela Saúde had no role in the study concept and design, data collection, analysis, interpretation, or writing of the manuscript. Grupo Fleury and IBOPE contributed to the funding of the SoroEpi MSP by providing their services at or below cost. CMTM received a scholarship (CNPq-Pq): 308974/2018-2; WVS CNPq-Pq: 308000/2021-8.

**Competing interests:** BHT, CMTM, MCGPA, FC, RTIB, EJAL, WVS, LCR, and FCR declare no conflict of interest. Authors associated with Grupo Fleury (CFHG, MCTP and EGR) and Ibope Inteligência (MCN) disclose the following potential conflict of interest: the two organizations co-funded the SoroEpi MSP Study by providing their services at or below cost. These include data and blood sample collection and laboratory tests. The companies sell these services in the market and might profit from the publicity generated by the results of this research. This does not alter our adherence to PLOS ONE policies on sharing data and materials.

## Results

The study sample comprised 8,134 adults. The overall SARS-CoV-2 seroprevalence increased from 11.4% (95%CI: 9.2–13.6) in June 2020 to 24.9% (95%CI: 21.0–28.7) in January 2021; from 38.1% (95%CI: 34.3–41.9) in April 2021 to 77.7% (95%CI: 74.4–81.0) in April 2022. The prevalence over time was higher in the subgroup 18–39 years old than in the older groups from Survey 3 onwards. The self-declared Black or mixed (*Pardo*) group showed a higher prevalence in all surveys compared to the White group. Monthly prevalence rose steeply from January 2021 onwards, particularly among those aged 60 years or older. The infection-to-case ratios ranged from 8.9 in June 2020 to 4.3 in January 2021.

## Conclusions

The overall seroprevalence rose significantly over time and with age and race subgroup variations. Increases in the 60 years or older age and the White groups were faster than in younger ages and Black or mixed (*Pardo*) race groups in the post-vaccination period. Our data may add to the understanding of the complex and changing population dynamics of the SARS-CoV-2 infection, including the impact of vaccination strategies and the modelling of future epidemiological scenarios.

## Introduction

Brazil was one of the countries most affected by the COVID-19 pandemic [1]. From February 2020 to 20 September 2022, approximately 34.6 million confirmed cases of COVID-19, with 685,000 deaths, were reported [2]. During that period, Brazil contributed to 5.6% of the global COVID-19 cases and around 10% of the total reported deaths [3], even though it accounts for only 2.3% of the world population [4].

São Paulo is among the top ten metropolitan areas of the world. It is in the wealthiest Brazilian region and has a high level of socioeconomic inequality [5]. The city of São Paulo, with a population of approximately 11.9 million [6], was the site of SARS-CoV-2 entry into the country and the initial epicenter of the pandemic. The city's population was an important route of viral spread [7, 8]. As of 14 September 2022, the São Paulo Municipality reported 2,281,679 confirmed cases and 43,785 deaths [9].

Cross-sectional surveys are snapshots of infection history. They can complement data based on PCR-confirmed cases by identifying asymptomatic infections and the magnitude of transmission in different settings and populations [10]. Repeated serosurveys can monitor the spread of the virus in the general population and describe the geographical and subgroup changes over time. Knowledge of trends of SARS-CoV-2 infections can inform and help evaluate the impact of public health actions to control the pandemic and shed light on the dynamics of the interaction between the virus and the population [11].

A systematic review of global SARS-CoV-2 seroprevalence from January 2020 to May 2022 indicated that most population-based studies were conducted before mass COVID-19 vaccination, and very few studies sampled populations in 2022 [12]. In Brazil, a nationwide household survey showed city-level prevalence had increased from 0% to 25.4% by June 2020, with a high heterogeneity among regions [13]. In the municipality of São Paulo, as part of the local public health response, sequential population-based household surveys estimated 9.7% SARS-CoV-2 seroprevalence in July 2020, with an increase to 25% by February 2021 [14]. SARS-CoV-2 has

been difficult to control and still imposes challenging uncertainties, and serial population-based studies help understand the infection's complex and changing population dynamics.

The objectives of this study were to describe the SARS-CoV-2 seroprevalence estimates in the municipality of São Paulo and to explore the seroprevalence trends by sociodemographic characteristics, considering pre- and post-vaccine implementation in the city. Lastly, we compared the overall seropositivity in the adult population with the COVID-19 cumulative registered cases during the pre-vaccination period, before January 17, 2021.

## Materials and methods

### Study design and sampling

The SoroEpi Study in the municipality of São Paulo (SoroEpi MSP) comprised seven household-SARS-CoV-2 serosurveys from June 2020 to April 2022: Survey 1 (15–24 June 2020), Survey 2 (20–29 July 2020), Survey 3 (1–10 October 2020), Survey 4 (14–23 January 2021), Survey 5 (22 April-1 May 2021), Survey 6 (9–20 September 2021), and Survey 7 (31 Mar-9 April 2022). The dates of the surveys were established according to the progress of the epidemic in São Paulo and to include the pre and post-vaccination periods. Fig 1 shows the number of households and adults by survey, which followed the study design.

Before initiating the full-scale surveys, a pilot study was conducted in May 2020, restricted to the six most COVID-19-affected districts in São Paulo. The data collection procedures proved to be adequate and feasible. The detailed methodology and preliminary seroprevalence of SARS-CoV-2 were published previously [15]. To provide the COVID-19 local epidemiological context for each survey time frame, secondary data were retrieved from the official sources of the reported cumulative cases and deaths (Fig 2). We also retrieved SARS-CoV-2 variants circulating during the period of surveys in São Paulo state, traced by GISAID [16]. From June 2020 to January 2021 the predominant circulating variant was B1.1 with the entry of the Gamma variant of concern (VOC) at the beginning of 2021. The Gamma variant was the main circulating VOC by April 2021; Delta VOC by September 2021 and Omicron VOC by April 2022.

The surveys were designed following the World Health Organization (WHO) protocol for population-level COVID-19 antibody testing [17]. They were cross-sectional population-based studies, and the target population was the adult (18 years old or older) inhabitants of permanent private households in the municipality of São Paulo.

According to the 2020 projections for the municipality of São Paulo by the Statewide System for Data Analysis Foundation (Fundação SEADE), 9,164,926 residents were 18 years of age and over out of an estimated total population of 11,869,660 [18]. Administratively, the municipality is organized into 96 districts. For this study, the adult population was divided into two strata of equal size using the average income of the municipality districts as the stratification variable. A sample size of 560 individuals was planned in each stratum to allow the estimation of seroprevalence greater than 4% to be obtained under coefficients of variation of less than 30%, considering a design effect of 2 (deff = 2).

We used a two-cluster probability sampling method to select the survey participants. In the first stage, we selected census tracts (115 in Survey 1 and 160 in the other six surveys) using probability proportional to the number of households counted in the 2010 census. In the second stage, we randomly selected households from each tract (12 in Survey 1 and eight in the subsequent surveys) from a list of households obtained by plotting out the tract. All adults in the selected households were eligible to participate in the study, except those with cognitive incapacities or health problems that make venipuncture difficult. Weights were introduced in the analysis to correct the different probabilities with which participants were randomly selected. These design weights were adjusted for non-response considering the response rates

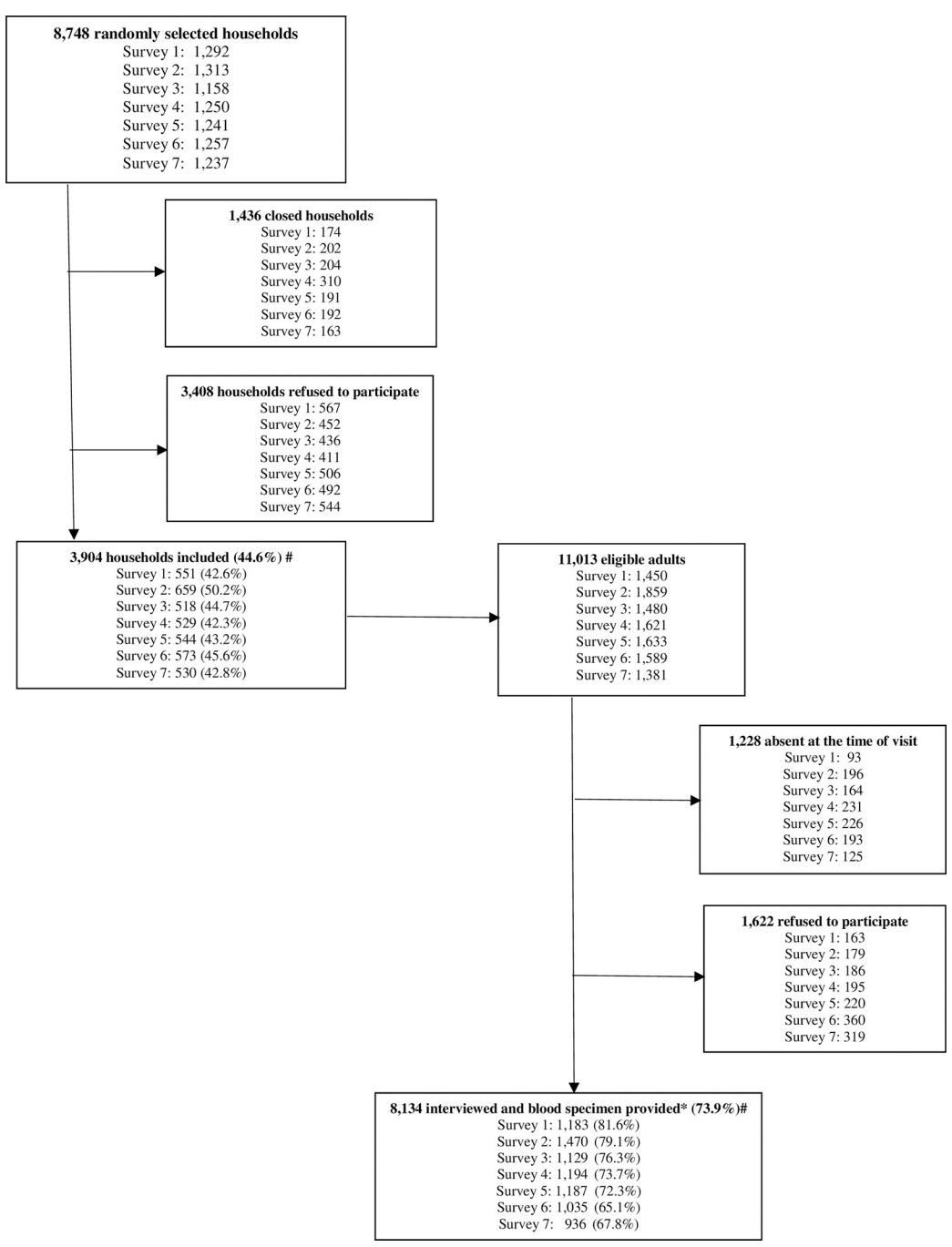

**Fig 1. Flow diagram for recruitment into the study by serosurvey, São Paulo, SP, Brazil, June 2020 to April 2022.** *19 individuals excluded due to lack of blood specimen: Survey 1: 11; Survey 2: 14; Surveys 3, 4, 6, 7: 1.

of three sets of census tracts (ordered by average income, according to census data) and for the age and sex distribution observed in the city of São Paulo in the years 2020, 2021, and 2022.

## Ethics and study procedures

Approval for the study protocol was obtained from the National Research Ethics Commission (CONEP) and the Research Ethics Committee of Fleury Group (CAAE no

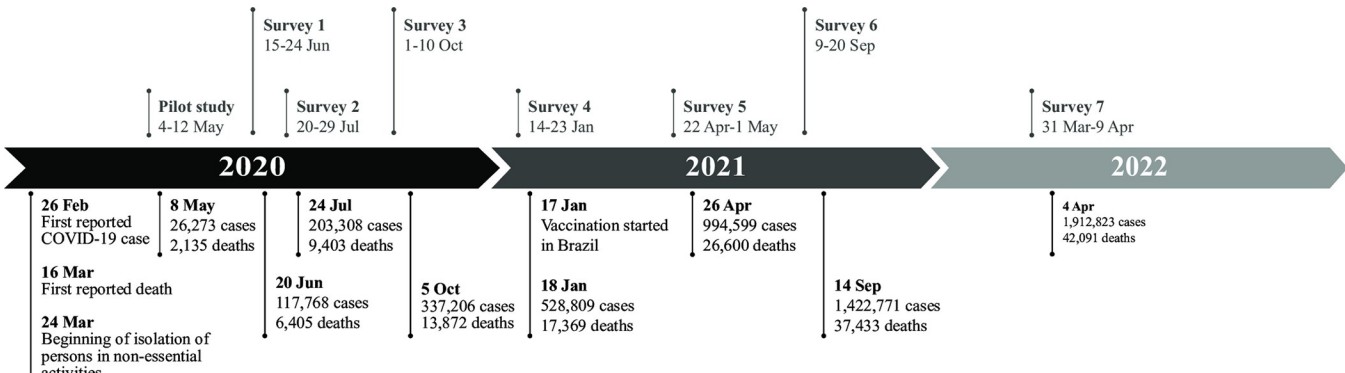

**Fig 2. Timeline of COVID-19 cases and deaths, SARS-CoV-2 variants by serosurveys, São Paulo, Brazil, 2020–2022.** SARS-CoV-2 predominant variants: B1.1 (June 2020-January 2021); Gamma (from 2021); Delta (September 2021) and Omicron (April 2022). Genomic surveillance data available from: https://www.genomahcov.fiocruz.br/dashboard-pt/ Reported cumulative COVID-19 cases and deaths available at: https://www.prefeitura.sp.gov.br/cidade/secretarias/saude/vigilancia_em_saude/index.php?p=295572.

31032620.0.0000.5474) on May 15, 2020. An amendment to the original serology survey was also approved by the Brazilian National Research Ethics Commission (CONEP) (CAAE no 31032620.0.0000.5474 v5) on January 14, 2021.

Three visits were scheduled for each household on different days of the week and times of the day to limit non-response. The sampling method did not permit replacements of the randomly selected sampling units, i.e., the census tracts and households. We recorded the frequency of households with eligible adults and the frequency of individuals interviewed with valid serological samples in each survey. We estimated the overall participation rate considering the number of included participants divided by the total number of individuals who would have been included according to the sampling frame, but who were considered non-participants because they were not reachable, or households could not be accessed. We assumed that the number of individuals per household was similar between the included households and the non-accessible selected households.

After participants signed the written informed consent, face-to-face interviews were conducted using a questionnaire on mobile devices to obtain socio-demographic and other data. The information used in the present study comprised: sex, age, skin color/race and education measured by years of formal schooling. Self-reported skin color/race categories followed the Brazilian Institute of Geography and Statistics (IBGE) classification: White, Black, mixed race (*Pardo*), Asian, and Indigenous [19]. Questions about the date of vaccination, the number of doses, and the type of vaccine product were included in the questionnaire to account for the introduction of vaccination during Surveys 5, 6, and 7. The COVID-19 immunization program commenced nationally on 17 January 2021, and on the same date in São Paulo city (Fig 2). In São Paulo State, four types of vaccines were used based on their availability and immunization policies. The first vaccines introduced in the early phase of the Brazilian COVID-19 campaign were Coronavac (Sinovac), an inactivated whole-virus vaccine, and Covishield (AstraZeneca). These were followed by Ad26.COV2.S (Janssen) and BNT162b2 (Pfizer-BioNTech) vaccines. However, not all vaccines were available throughout the vaccination campaign due to shortages and delays [20].

Following the interview, about 3–5 ml of venous blood was collected from each participant, and samples were transported to the reference laboratory for antibody testing. The laboratory team had access to information on individual participants during and after data collection to enable them to provide details of their results and any additional information the participants needed.

## Serology testing

Specimens were tested for anti-nucleocapsid total antibodies using the electrochemilumines-cence Roche Elecsys Total Ig Assay (Roche Elecsys® Diagnostics, Rotkreuz, Switzerland) in Surveys 2 to 7. The sensitivity of the Elecsys Anti-SARS-CoV-2 immunoassay is 99.5% (95% CI 97–100) and specificity is 99.8% (95% CI 99.7–99.9), making it a useful tool for determining past SARS-CoV-2 infection, including at population level [21]. For Survey 1, the specimens were tested for immunoglobulin (Ig) M (IgM) and IgG using Maglumi SARS-CoV-2 chemilu-minescence assay Immunoglobulin (Snibe Diagnostics, Shenzhen, China, with a sensitivity of 100% and a specificity of 94.1% for IgM, and a sensitivity of 100% and a specificity of 99.5% for IgG [22]. These were the available tests in the early months of the COVID-19 pandemic in 2020. ANVISA, the Brazilian Regulatory Health Agency, has cleared both assays for emergency use. Other tests were used in parallel with the Elecsys assay during the study period. However, for the sake of the present analyses, we opted to measure the anti-nucleocapsid total antibodies using Roche Elecsys due to its reported high sensitivity and specificity and because it has been broadly used in seroprevalence studies [21, 23]. All tests were performed following the manu-facturer's guidelines in the Grupo Fleury laboratory in São Paulo, Brazil.

## Primary outcome

The primary outcome was seropositivity, which was defined according to previously set labo-ratory parameters. Individuals whose sera were reactive to the Maglumi assay in Survey 1 or Roche assay in Surveys 2 to 7 were considered positive. Indeterminate results were included as negative in the analyses. After the CoronaVac vaccine was introduced in early 2021, seroposi-tivity could be determined either by a viral infection or by vaccination and herein is defined as "hybrid immunity" [24].

## Statistical analysis

Unadjusted and weighted frequency distributions were calculated with 95% confidence inter-vals (CI) by sex, age group, self-reported skin color, schooling, and mean income study design strata. Differences between categories were assessed using Rao-Scott chi-square tests. Estimates were obtained using the survey module of STATA version 14 (StataCorp, College Station, Texas, USA), in which the complex aspects of the sampling design were considered.

We applied joinpoint regression analysis to study the trends of sequential seroprevalence estimates of SARS-CoV-2 antibody surveys from June 2020 to April 2022 (seven surveys) using the JoinPoint app [25]. This model allowed us to test whether the changes in the sero-prevalence trend were statistically significant and to identify the time of the turning point sepa-rating the trend into distinct segments. Our dataset allowed us to choose one as the maximum number of joinpoints in the linear model $y = \alpha + \beta x$. The model selection used Monte Carlo permutation tests to determine the optimal number of joinpoints [26]. Each one performs a test of the null hypothesis $H_0$: 0 joinpoint against the alternative $H_a$: 1 joinpoint. This time series approach can distinguish different growth trends and their period (trend segments) according to the variables analysed. Therefore, the location of the joinpoint (period) was based on statistical criteria resulting from the joinpoint calculations.

## Calculation of the reported fraction

We calculated the reported fraction as the ratio of the estimated number of infected individuals using primary seroprevalence data (Surveys 1 to 4) and the number of cumulative cases of COVID-19 extracted from the São Paulo Municipal Surveillance System. We considered the

median time of the corresponding fieldwork period. The reported fractions were calculated for the surveys performed before the mass CoronaVac vaccination was introduced in São Paulo.

## Results

Fig 1 shows the number of randomly selected households for each survey with a total of 8,748. In 3,904 (44.6%) households, eligible adults were identified, at 1,436 (16.4%) there was no answer, and in 3,408 (39.0%) households, their residents refused to give any information.

The seven consecutive surveys included 8,134 participants with complete interviews and blood specimens, which corresponded to 73.9% of the 11,013 eligible adults. However, if we consider the participation rate of the selected households together with the participation rate of the eligible adults, the overall participation rate was 33.0% (Fig 1).

For each survey, the percentages of participants by sex, age group, and mean income study design strata agreed with the percentages of the population of the municipality of São Paulo after the sampling design weights were adjusted for these variables. Regarding race, approximately 50% of the participants self-declared as White, followed by approximately 42% as Black or mixed race (*Pardo*), and less than 5% as Asian or Indigenous. The percentage of participants with 11 years or less of schooling ranged from 32.9% (Survey 2) to 39.5% (Survey 7) (S1 Table).

From June 2020 (Survey 1) to January 2021 (Survey 4), before mass vaccination, the weighted overall seroprevalence increased from 11.4% (95% CI: 9.2–13.6) to 24.9% (95% CI: 21.0–28.7). From April 2021 (Survey 5) to April 2022 (Survey 7), the seropositivity increased from 38.1% (95% CI: 34.3–41.9) to 77.7% (95% CI: 74.4–81.0). Residents in the lower-income stratum had a statistically higher weighted prevalence of SARS-CoV-2 antibodies than the high-income stratum in all surveys except for Survey 2. None of the surveys showed a significant difference in the frequency of seropositivity between women and men. Among the oldest age group ($\geq$ 60 years), the prevalence ranged from 11.1% (95% CI: 6.0–16.2) in Survey 1 to 65.4% (95% CI: 58.1–72.6) in Survey 7, while in the youngest age group (18–39 years), it varied from 10% (95% CI: 7.1–12.8) to 85.8% (95% CI: 81.6–90.0). The self-declared Black or mixed race (*Pardo*) group presented statistically higher seroprevalence in all surveys when compared to the White group. The Asian or Indigenous group did not provide reliable estimates due to its small sample size and large 95% confidence intervals. The population with 16 or more years of education had statistically significant lower seroprevalence in all surveys (Table 1).

In April 2021 (Survey 5), 83.7% (95% CI: 81.1–85.9) of the São Paulo city population were unvaccinated, and 10.5% (95% CI: 8.5–12.9) had received at least one dose of CoronaVac. There was a sharp decline in the percentage of unvaccinated adults to 4.0% (95% CI: 2.8–5.8) in September 2021 (Survey 6) and 1.8% (95% CI: 0.9–3.5) by April 2022 (Survey 7). The CoronaVac vaccine (one or more doses) was the most commonly used in the oldest age group (60 years or older) and the 18–39 years age group (S2 Table).

Table 2 shows the growth model parameters for prevalence stratified by sex, age group and self-reported race/skin color, and Fig 3 presents the observed and modelled prevalence data for each of these variables. The overall trend of the point prevalence series (Fig 3A) identified two periods with significantly different growths: a monthly increase of approximately 2% ($\beta1$) in prevalence from June 2020 (Survey 1) to January 2021 (Survey 4) and a monthly increase of approximately 3.5% ($\beta2$) in prevalence from January 2021 (Survey 4) to April 2022 (Survey 7). For both sexes (Fig 3B), there is a joinpoint in January 2021 (Survey 4), and the overlap of the prevalence curves in the whole study period illustrates the lack of difference between the two groups. The trend analysis of the prevalence by age group (Fig 3C) indicated one joinpoint in January 2021 (Survey 4) for the 18–39 and 60 or older groups. After this point, both groups

**Table 1. Weighted seroprevalence estimates[a] (and 95% CI) of anti-SARS-CoV-2 per survey.** The SoroEpi MSP Study, Municipality of São Paulo, SP, Brazil, June 2020 to April 2022.

| Characteristic | Survey 1[b] June 2020 | | Survey 2 July 2020 | | Survey 3 October 2020 | | Survey 4 January 2021 | | Survey 5 April 2021 | | Survey 6 September 2021 | | Survey 7 April 2022 | |
|---|---|---|---|---|---|---|---|---|---|---|---|---|---|---|
| | n | % 95% CI | n | % 95% CI | n | % 95% CI | n | % 95% CI | n | % 95% CI | n | % 95% CI | n | % 95% CI |
| **Overall** | 1183 | 11.4 (9.2–13.6) | 1470 | 14.8 (12.1–17.6) | 1129 | 18.6 (15.2–22.0) | 1194 | 24.9 (21.0–28.7) | 1187 | 38.1 (34.3–41.9) | 1035 | 52.7 (48.7–56.8) | 936 | 77.7 (74.4–81.0) |
| **Study strata (mean income)** | p<0.0001 | | p = 0.2072 | | p = 0.0083 | | p = 0.0004 | | p = 0.0022 | | p = 0.0001 | | p<0.0001 | |
| High | 676 | 6.5 (4.4–8.5) | 851 | 13.0 (9.2–16.8) | 544 | 13.9 (9.6–18.2) | 580 | 17.6 (12.7–22.6) | 610 | 32.0 (26.7–37.4) | 415 | 44.9 (39.2–50.5) | 435 | 70.7 (65.1–76.2) |
| Low | 507 | 16.0 (12.2–19.8) | 619 | 16.5 (12.6–20.4) | 585 | 23.0 (18.0–28.0) | 614 | 31.9 (26.1–37.6) | 577 | 44.0 (38.8–49.2) | 620 | 60.4 (55.3–65.4) | 501 | 84.2 (80.8–87.6) |
| **Sex** | p = 0.8192 | | p = 0.7207 | | p = 0.6261 | | p = 0.8207 | | p = 0.3694 | | p = 0.6701 | | p = 0.8619 | |
| Male | 521 | 11.6 (8.7–14.5) | 607 | 14.5 (10.9–18.0) | 460 | 18.0 (13.2–22.8) | 483 | 24.6 (19.4–29.7) | 481 | 36.7 (31.7–41.7) | 368 | 53.7 (46.9–60.5) | 303 | 78.0 (73.3–82.6) |
| Female | 662 | 11.2 (8.5–13.9) | 863 | 15.1 (12.0–18.3) | 669 | 19.2 (15.7–22.6) | 711 | 25.1 (21.2–29.1) | 706 | 39.4 (34.7–44.0) | 667 | 51.9 (47.0–56.8) | 633 | 77.5 (73.5–81.5) |
| **Age group (years)** | p = 0.3585 | | p = 0.0850 | | p = 0.0237 | | p<0.0001 | | p = 0.0001 | | p = 0.0087 | | p<0.0001 | |
| 18–39 | 496 | 10.0 (7.1–12.8) | 613 | 15.0 (11.7–18.4) | 459 | 21.8 (17.2–26.4) | 497 | 29.4 (24.4–34.3) | 502 | 41.6 (36.3–47.0) | 414 | 58.8 (53.3–64.3) | 325 | 85.8 (81.6–90.0) |
| 40–59 | 435 | 13.4 (9.7–17.1) | 494 | 16.8 (12.3–21.2) | 397 | 18.2 (13.2–23.3) | 425 | 25.6 (20.6–30.6) | 462 | 40.7 (35.6–45.8) | 357 | 48.1 (41.7–54.4) | 360 | 75.3 (69.7–80.9) |
| ≥ 60 | 252 | 11.1 (6.0–16.2) | 363 | 10.9 (7.4–14.5) | 273 | 12.3 (7.5–17.2) | 272 | 14.1 (9.3–19.0) | 223 | 26.3 (20.5–32.1) | 264 | 47.9 (40.7–55.1) | 251 | 65.4 (58.1–72.6) |
| **Self-reported race/ skin color[c]** | p = 0.0055 | | p = 0.0543 | | p = 0.0012 | | p<0.0001 | | p<0.0001 | | p = 0.0011 | | p = 0.0051 | |
| White | 614 | 7.9 (5.6–10.4) | 733 | 12.9 (9.6–16.2) | 556 | 14.4 (10.6–18.2) | 609 | 18.0 (13.9–22.0) | 537 | 30.9 (25.9–35.9) | 406 | 45.3 (39.5–51.1) | 435 | 72.1 (67.1–77.0) |
| Black or mixed (*Pardo*) | 510 | 15.3 (11.9–18.7) | 671 | 17.0 (13.2–20.7) | 530 | 23.4 (19.0–27.8) | 549 | 33.3 (27.9–38.7) | 584 | 45.5 (40.8–50.2) | 599 | 58.6 (53.9–63.3) | 470 | 82.8 (79.0–86.5) |
| Asian or Indigenous | 48 | 9.7 (0.0–20.6) | 49 | 8.5 (1.6–15.4) | 33 | 11.8 (0.6–23.0) | 30 | 9.3 (0.0–19.9) | 58 | 28.7 (15.9–41.5) | 25 | 45.5 (21.6–69.3) | 24 | 80.4 (60.5–100) |
| **Schooling (years)** | p = 0.0004 | | p = 0.0007 | | p = 0.0001 | | p<0.0001 | | p<0.0001 | | p = 0.0231 | | p = 0.0469 | |
| ≤ 11 | 386 | 17.2 (12.2–22.2) | 518 | 17.8 (13.9–21.7) | 429 | 24.5 (18.7–30.3) | 442 | 29.7 (23.6–35.9) | 415 | 41.2 (35.4–47.0) | 417 | 55.1 (49.9–60.3) | 370 | 80.6 (76.5–84.7) |
| 12–15 | 464 | 11.0 (8.4–13.5) | 586 | 16.0 (12.2–19.8) | 426 | 19.0 (14.6–23.4) | 474 | 27.2 (22.7–31.7) | 483 | 46.1 (40.7–51.6) | 416 | 56.4 (51.1–61.7) | 381 | 78.7 (73.8–83.6) |
| ≥ 16 | 333 | 5.1 (1.8–8.4) | 366 | 8.9 (5.9–11.9) | 274 | 9.7 (6.0–13.4) | 278 | 14.3 (9.7–18.9) | 289 | 21.5 (15.6–27.4) | 202 | 42.8 (33.0–52.6) | 185 | (62.5–78.3) |

[a]In all surveys, seroprevalence estimates were weighted by sampling design with adjustments for non-response.

[b]Survey 1, the specimens were tested for IgM and IgG with the Maglumi SARS-CoV-2 test.

In Surveys 2 to Survey 7, the specimens were tested for anti-nucleocapsid total antibodies with the Roche Elecsys test.

[c]Missing data were: 11, 17, 10, 6, 8, 5 and 7 in the Surveys 1 to 7, respectively

presented similar monthly increases (β2 = 3.5% and 3.7%, respectively), although the prevalence magnitudes differed. For the White self-reported race group, the set identified one joinpoint in January 2021 (Survey 4) and a significant increase of 3.5% between January 2021 and April 2022, whereas the Black or mixed self-reported race group showed a significant monthly increase of 3.1% over the whole study period (Fig 3D). Asian or Indigenous population were not included in this analysis because the prevalences could not be interpreted due to small sample size.

**Table 2. Growth models and parameter estimates of point prevalence of SARS-CoV-2 antibody encompassing seven population-based surveys.** The SoroEpi MSP Study, Municipality of São Paulo, SP, Brazil, June 2020 to April 2022.

| | Period[a] | Parameter | Estimate | Standard Error | Test Statistic (t) | Prob > |t| |
|---|---|---|---|---|---|---|
| **Overall** | | | | | | |
| | Jun 2020—Jan 2021 | α1 | 11.785 | 0.838 | 14.065 | 0.005 |
| | | β1 | 1.987* | 0.445 | 4.463 | 0.047 |
| | Jan 2021—Apr 2022 | α2 | 1.404 | 3.130 | 0.448 | 0.698 |
| | | β2 | 3.470* | 0.183 | 18.958 | 0.003 |
| **Study strata (mean income)** | | | | | | |
| High | Jun 2020—Apr 2022 | α1 | 6.093 | 1.686 | 3.614 | 0.015 |
| | | β1 | 2.702* | 0.221 | 12.239 | 0.000 |
| Low | Jun 2020—Oct 2020 | α1 | 15.475 | 0.614 | 25.217 | 0.002 |
| | | β1 | 1.792 | 0.877 | 2.043 | 0.178 |
| | Oct 2020 –Apr 2022 | α2 | 8.947 | 1.050 | 8.518 | 0.014 |
| | | β2 | 3.424* | 0.062 | 55.399 | 0.000 |
| **Sex** | | | | | | |
| Male | Jun 2020—Jan 2021 | α1 | 11.849 | 0.519 | 22.849 | 0.002 |
| | | β1 | 1.883* | 0.294 | 6.415 | 0.023 |
| | Jan 2021- Apr 2022 | α2 | 0.210 | 1.999 | 0.105 | 0.926 |
| | | β2 | 3.546* | 0.117 | 30.191 | 0.001 |
| Female | Jun 2020—Jan 2021 | α1 | 11.815 | 1.229 | 9.611 | 0.011 |
| | | β1 | 2.056 | 0.586 | 3.508 | 0.073 |
| | Jan 2021—Apr 2022 | α2 | 2.337 | 4.713 | 0.496 | 0.669 |
| | | β2 | 3.410* | 0.273 | 12.469 | 0.006 |
| **Age group (years)** | | | | | | |
| 18–39 | Jun 2020—Jan 2021 | α1 | 10.895 | 0.931 | 11.704 | 0.007 |
| | | β1 | 2.726* | 0.522 | 5.219 | 0.035 |
| | Jan 2021—Apr 2022 | α2 | 4.006 | 3.718 | 1.078 | 0.394 |
| | | β2 | 3.710* | 0.212 | 17.502 | 0.003 |
| 40–59 | Jun 2020—Apr 2022 | α1 | 11.563 | 1.998 | 5.789 | 0.002 |
| | | β1 | 2.725* | 0.208 | 13.087 | 0.000 |
| ≥ 60 | Jun 2020—Jan 2021 | α1 | 10.458 | 1.593 | 6.563 | 0.022 |
| | | β1 | 0.652 | 0.733 | 0.890 | 0.467 |
| | Jan 2021—Apr 2022 | α2 | -9.737 | 5.423 | -1.795 | 0.214 |
| | | β2 | 3.537* | 0.348 | 10.170 | 0.010 |

[a]The JoinPoint analysis identified the number of turning points and period (trend segment) for each variable.

*Slope significantly different from zero at the alpha = 0.05 level.

Table 3 shows the estimated population who had had SARS-CoV-2 infection in the pre-vaccination period (Surveys 1 to 4) in the municipality of São Paulo, the number of cumulative reported COVID-19 cases and the infection-to-case ratio. The estimated number of persons who were infected ranged from 1.1 million in June 2020 (Survey 1) to approximately 2.3 million in January 2021 (Survey 4). The underreporting multiplier varied from 8.9 in Survey 1 to 4.3 in Survey 4.

## Discussion

The present study presents new data on the seroprevalence trends of SARS-CoV-2 antibodies in the general adult population of the municipality of São Paulo, Brazil, based on seven cross-

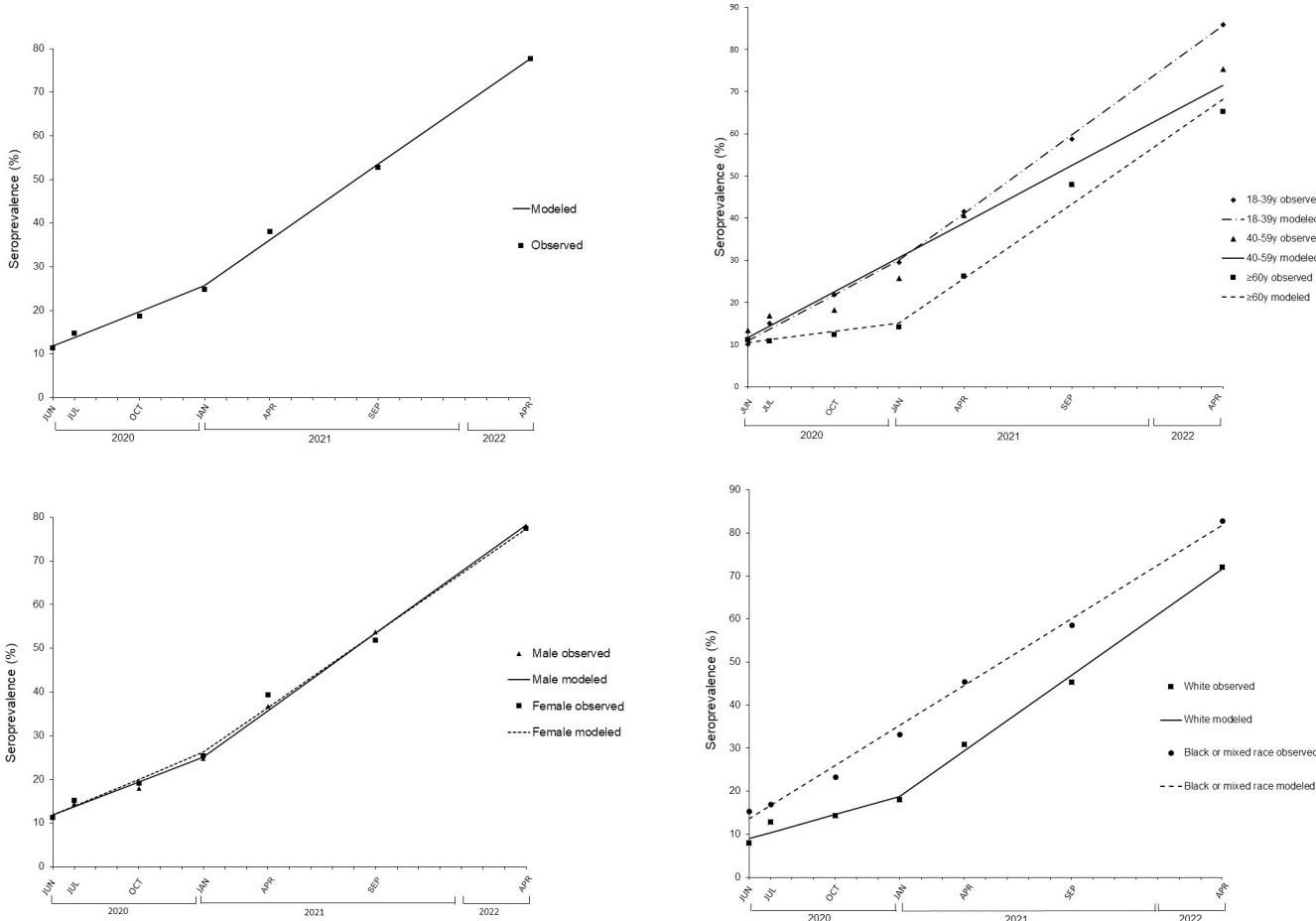

**Fig 3. Joinpoint trends for SARS-CoV-2 seropositivity in São Paulo city, June 2020 to April 2022.** a) overall, b) sex, c) age group, and d) White and Black or mixed (Pardo) skin color.

**Table 3. Estimated number of adults infected with SARS-CoV-2 and the ratio of estimated infections by reported COVID-19 cases in the pre-vaccination period.** The SoroEpi MSP Study, Municipality of São Paulo, SP, Brazil, June 2020 to January 2021.

| Survey (specimen collection period) | Populationª | Sample size n | SARS-CoV-2 seroprevalence % (95% CI) | Estimated number of adults with SARS-CoV-2 infection n (95% CI) | Cumulative reported COVID-19 cases (date)ᵇ | Ratio (infections: cases) |
|---|---|---|---|---|---|---|
| Survey 1 (15–24 June 2020) | 9,164,926 | 1,183 | 11.4 (9.2–13.6) | 1,044,802 (843,173–1,246,430) | 117,768 (20 June) | 8.9 (7.2–10.6) |
| Survey 2 (20–29 July 2020) | 9,164,926 | 1,470 | 14.8 (12.1–17.6) | 1,356,409 (1,108,956–1,613,027) | 203,308 (24 July) | 6.7 (5.5–7.9) |
| Survey 3 (1–10 October 2020) | 9,164,926 | 1,129 | 18.6 (15.2–22.0) | 1,704,676 (1,393,069–2,016,284) | 337,206 (5 October) | 5.1 (4.1–6.0) |
| Survey 4 (14–23 January 2021) | 9,216,158 | 1,194 | 24.9 (21.0–28.7) | 2,294,823 (1,935,393–2,645,037) | 528,809 (18 January) | 4.3 (3.7–5.0) |

ªPopulation aged 18 years and older projected by Fundação SEADE for São Paulo City.

ᵇSource: Secretaria Municipal de Saúde, São Paulo.

sectional surveys over an almost 2-year period. Our data showed that SARS-CoV-2 seropositivity increased over time, ranging from 11.4% (95%CI: 9.2–13.6) in June 2020 to 77.7% (95% CI: 74.4–81.0) in April 2022. We observed a significantly higher increase in the seroprevalence trend from January 2021 to April 2022 compared to 2020, most probably driven by the implementation of the COVID-19 vaccination campaigns.

Our results revealed that approximately 80% of the adult population in the municipality of São Paulo was susceptible to SARS-CoV-2 infection before vaccine availability in January 2021. A relatively small variation, from 11.4% (95%CI: 9.2–13.6) in June 2020 to 18.6% (95% CI: 15.2–22.0) in October 2020, measured natural immunity using an anti-nucleocapsid antibody test. In the pre-COVID-19 vaccine era, serosurveys using rapid tests (lateral flow Wondfo Biotech) in the city of São Paulo showed an even lower seroprevalence of SARS-CoV-2 (13.6%) by September 2020 [14]. A systematic review and pooled analysis estimated that around 10% of the Brazilian population had been infected by March 2021 [27]. Globally, there was a large variation in prevalence among the geographic regions over time, and the authors called attention to the need for continuous monitoring of the population's immunity levels [12, 27, 28].

Buss et al reported 44% of prevalence of IgG antibodies to SARS-CoV-2 nucleocapsid (N) protein (Abbot) among blood donors in the city of Manaus (Brazilian Amazon) by June 2020 [29]. The prevalence rate was higher in the Amazon than in São Paulo (13.6%, 95% CI: 12.0–18.1) in the same period, showing distinct patterns of the epidemic within the country [29]. The prevalence reported among blood donors is similar to the 11.4% (95% CI: 9.2–13.6) found in the general population in June 2020 (Survey-1). A serial cross-sectional SARS-CoV-2 antibody (Abbot assay) among blood donors showed a range of adjusted seroprevalence from 19.0% (95% CI 15.9–22.4) in December 2020 to 22.3% (95%CI19.0–25.9) in February 2021 in the municipality of São Paulo [30]. These findings are in line with our population-based level of 24.9% (95% CI 21.0–28.7) of weighted seroprevalence of anti-SARS-CoV-2 in January 2021 (Survey-4), which is within the time period analysed by Prete et al [30]. Serosurveys conducted on convenience samples of blood donors are prone to selection population bias, but their results may be considered a proxy measure of the SARS-CoV-2 seroprevalence in the adult population. Besides the difference in the study populations, comparisons of our results with blood donor studies may be made with caution because of differences in the assays performed.

The high level of seropositivity in the adult population was only reached by the date of our last survey (April 2022). This population-level immunity reflected the past virus infection and/or vaccination, mostly with CoronaVac, eliciting anti-nucleocapsid antibodies. The high level of hybrid immunity was mainly achieved by extensive vaccination coverage. We acknowledge that the reported positivity of SARS-CoV-2 antibodies at the population level did not include the anti-Spike antibodies elicited by other vaccine types [31]. In this sense, we may assume that the overall seroprevalence in our study could be underestimated. A seroprevalence of more than 95%, using SARS-CoV-2 anti-Spike protein assay, was reported among blood donors in the city of São Paulo and six other Brazilian capitals in November 2021 [32]. Our results were similar to those observed in serial surveys among blood donors in the US, where the seroprevalence induced by hybrid immunity reached more than 80% [33]. In our study, adults living in districts classified as low-income stratum, and those who self-declared Black or mixed (*Pardo*), were the most infected over the entire study period, in line with other serosurveys conducted in Brazil [14, 34] and elsewhere [27]. In our setting, during 2020, higher COVID-19 mortality was observed among the lower-income stratum and Black populations, showing marked social inequalities [35, 36].

In our models, the upward trend of the overall seroprevalence and the seroprevalence estimates stratified by sex, age group and White and Black or mixed (*Pardo*) skin color indicated

that January 2021 was the inflection point in the study period. The overlapping of the growth curves for males and females suggested similarities in the frequency of SARS-CoV-2 viral infection as the result of natural immunity in the year 2020 and/or vaccine-induced immunity afterwards. People over 60 years of age showed low seroprevalence, around 10%, with no significant increase in the year 2020, in contrast to significant increases among the younger age groups. This likely reflected the higher compliance to lockdown or social distance measures among the older population reported in other Brazilian regions [37]. Considering the fitted models for those aged 60 years or older, there was around a threefold increase in the slope estimated after January 2021. This high slope in prevalence growth occurred concurrently with the prioritization of COVID-19 vaccination among the older population. Therefore, vaccination seems to be the likely explanation for the rapid increase in hybrid immunity in older individuals. Of note, approximately half of the older population received the CoronaVac vaccine, the first vaccine product available in Brazil [20].

We found at least 8.9 (June 2020) and 4.3 (January 2021) times higher numbers of SARS-CoV-2 infections than the number of COVID-19 cases reported by the official surveillance system in 2020. These figures aligned with the 6.9 overall estimated ratio of the SARS-CoV-2 infection by the confirmed COVID-19 reported cases in the region of the Americas, based mainly on US serosurveys [38]. Several possible explanations could be offered for the variation of the infections:reported cases ratio in our setting. Asymptomatic or mild symptomatic cases were seldom reported; laboratory tests for disease confirmation were restricted to severe cases in the early phases of the epidemic; and there were changes in the criteria for reporting COVID-19 cases, with cases being confirmed based on symptoms and imaging tests [39].

Population-based seroprevalence surveys of SARS-CoV-2 can provide estimates of the magnitude of SARS-CoV-2 infections irrespective of case-based surveillance [40]. In the UK, the Office for National Statistics has been performing repeated antibody and RT-PCR swab surveys since 2020, in addition to gathering case surveillance data [41]. Also, in the US, a large national probabilistic repeated household survey has been in place to estimate the cumulative incidence in the stages of the COVID-19 pandemic [42].

SARS-CoV-2 was difficult to control and still imposes challenging uncertainties [43]. Sequential population-based studies with probabilistic sampling and highly accurate laboratory assays are central to monitoring the infection's complex and changing population dynamics.

To our knowledge, this was the only seven-round study of the point prevalence of SARS-CoV-2 antibodies in a representative sample of the general adult population of the municipality of São Paulo. It included the pre- and post-vaccination periods throughout the two years of the pandemic. A key strength of our study was the use of a robust methodology including the probability sampling method to select the survey participants, together with statistical weighting and adjustments for demographics and non-response. The use of a highly sensitive, specific and fully automated assay to detect anti-nucleocapsid SARS-CoV-2 immunoglobulins was another strength. Moreover, the same rigorous methodology, data analysis strategies, and testing procedures were repeated in all surveys, despite the adverse conditions such as periods of lockdown and social distancing.

Although our study has significant strengths, several limitations must be acknowledged. The first limitation was that we did not include people under 18 years old due to logistic, time, and cost restrictions. Second, like most population-based household serosurveys [12], our study had a low response rate. This fact may have introduced bias related to the reasons why different groups did not agree to be tested throughout the study period. Nevertheless, there is a consistent upward trend of seroprevalence in all categories within the monitoring period. Third, we used an anti-SARS-CoV-2-nucleocapsid test that did not distinguish between

antibodies induced by natural infection and those induced by vaccines. Fourth, seroprevalence results in Survey 1 were derived from the available laboratory test at the time, which was not the same as the test used in Surveys 2–7. Since there was only a one-month interval between Survey 1 and Survey 2, and both yielded similar estimates, we judged that it did not introduce measurement bias. Fifth, we did not consider the antibody waning over the months following infection, which could have led to underestimated results. However, we used Roche Elecsys immunoassay, which is considered a high sensitivity, high specificity assay and less prone to antibody waning [44]. Our study design which consisted of seven independent population-based surveys did not allow us to measure antibody waning directly as in the cohort studies. Also, the time of natural infection or the vaccine intake in relation to the blood collection was out of the scope of our study. Last, it is important to recognize that the joinpoint trend analysis is usually performed in longer series of point estimates composed of higher numbers of observations.

## Conclusion

Our results documented the evolution of the SARS-CoV-2 infection patterns and vaccine response throughout the two years of the pandemic in the municipality of São Paulo. It enabled us to learn about the epidemiological features of the infection, evidencing the complexity of the population-based seropositivity profile. Our study reinforced the importance of conducting repeated serological surveillance to monitor and study how new infectious pathogen transmission dynamics evolve in susceptible populations. Such data may add to the understanding of the complex and changing population dynamics of the SARS-CoV-2 infection, including the impact of vaccination strategies and the modelling of future epidemiological scenarios.

## Supporting information

**S1 Table. Frequency of selected characteristics of the sample population per survey.** The SoroEpi MSP Study, Municipality of São Paulo, SP, Brazil, June 2020 to April 2022.
(DOCX)

**S2 Table. Frequency of adults (and 95% Confidence Interval) by type of anti-SARS-CoV-2 vaccine received and age group in Surveys 5, 6 and 7.** The SoroEpi MSP Study, Municipality of São Paulo, SP, Brazil, June 2020 to April 2022.
(DOCX)

## Acknowledgments

We thank all the study participants whose cooperation made the study possible.

## Author Contributions

**Conceptualization:** Beatriz Helena Tess, Maria Cecília Goi Porto Alves, Marcia Cavallari Nunes, Fernando de Castro Reinach, Celso Francisco Hernandes Granato, Maria Carolina Tostes Pintão.

**Data curation:** Maria Cecília Goi Porto Alves, Regina Tomie Ivata Bernal.

**Formal analysis:** Celina Maria Turchi Martelli, Maria Cecília Goi Porto Alves, Regina Tomie Ivata Bernal, Wayner Vieira de Souza.

**Funding acquisition:** Fernando de Castro Reinach.

**Investigation:** Beatriz Helena Tess, Maria Cecília Goi Porto Alves, Marcia Cavallari Nunes, Fernando de Castro Reinach, Celso Francisco Hernandes Granato, Maria Carolina Tostes Pintão.

**Methodology:** Beatriz Helena Tess, Celina Maria Turchi Martelli, Maria Cecília Goi Porto Alves, Wayner Vieira de Souza, Marcia Cavallari Nunes, Fernando de Castro Reinach, Celso Francisco Hernandes Granato, Maria Carolina Tostes Pintão.

**Project administration:** Beatriz Helena Tess, Marcia Cavallari Nunes, Fernando de Castro Reinach, Maria Carolina Tostes Pintão.

**Resources:** Marcia Cavallari Nunes, Fernando de Castro Reinach, Celso Francisco Hernandes Granato, Maria Carolina Tostes Pintão.

**Supervision:** Marcia Cavallari Nunes, Fernando de Castro Reinach, Maria Carolina Tostes Pintão.

**Validation:** Celina Maria Turchi Martelli, Maria Cecília Goi Porto Alves.

**Visualization:** Celina Maria Turchi Martelli, Maria Cecília Goi Porto Alves, Fanny Cortes, Wayner Vieira de Souza.

**Writing – original draft:** Celina Maria Turchi Martelli, Fanny Cortes.

**Writing – review & editing:** Beatriz Helena Tess, Celina Maria Turchi Martelli, Maria Cecília Goi Porto Alves, Regina Tomie Ivata Bernal, Wayner Vieira de Souza, Expedito José de Albuquerque Luna, Laura da Cunha Rodrigues, Marcia Cavallari Nunes, Fernando de Castro Reinach, Celso Francisco Hernandes Granato, Edgar Gil Rizzatti, Maria Carolina Tostes Pintão.

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
