## [Decision Letter · Decision Letter 0]

5 Sep 2023

PONE-D-23-14879Seroprevalence trends of anti-SARS-CoV-2 antibodies in the adult population of the São Paulo Municipality, Brazil: Results from seven serosurveys from June 2020 to April 2022. The SoroEpi MSP StudyPLOS ONE

Dear Dr. TESS,

Thank you for submitting your manuscript to PLOS ONE. After careful consideration, we feel that it has merit but does not fully meet PLOS ONE’s publication criteria as it currently stands. Therefore, we invite you to submit a revised version of the manuscript that addresses the points raised during the review process.

We look forward to receiving your revised manuscript.

Kind regards,

Marília Jesus Batista de Brito Mota, Post-doc

Academic Editor

PLOS ONE

Journal Requirements:

"I have read the journal's policy and the authors of this manuscript have the following competing interests: authors associated with Grupo Fleury (CFHG, MCTP and EGR) and Ibope Inteligência (MCN) disclose the following potential conflict of interest: the two organizations co-funded the SoroEpi MSP Study by providing their services at or below cost. These include data and blood sample collection and laboratory tests. The companies sell these services in the market and might profit from the publicity generated by the results of this research.

BHT, CMTM, MCGPA, FC, RTIB, EJAL, WVS, LCR, and FCR declare no conflict of interest"

Additional Editor Comments:

This study presents an important contribution to the literature. Its major strength is to look at SARS-CoV-2 antibodies in a representative sample of the population, and to estimate underreporting using infection-case calculations.

The authors should considere several important studies that have looked at (pre-vaccination) SARS-CoV-2 antibodies in Sao Paulo. Namely, the authors need to compare their findings to the studies from Buss et al Science 2021 and Prete et al. eLife 2022, which looked at SARS-CoV-2 antibodies in (more cost effective) blood donor populations in Sao Paulo city using Abbot assays. Another important limitation is the lack of adjustments regarding sensitivity and specificity and seroreversion (if any) for the Roche assay.

Reviewer 1:

This study presents an important contribution to the literature. Its major strength is to look at SARS-CoV-2 antibodies in a representative sample of the population, and to estimate underreporting using infection-case calculations.

My main concern with this study is the omission of several important studies that have looked at (pre-vaccination) SARS-CoV-2 antibodies in Sao Paulo. Namely, the authors need to compare their findings to the studies from Buss et al Science 2021 and Prete et al. eLife 2022, which looked at SARS-CoV-2 antibodies in (more cost effective) blood donor populations in Sao Paulo city using Abbot assays. Another important limitation is the lack of adjustments regarding sensitivity and specificity and seroreversion (if any) for the Roche assay.

Other minor edits are also required before publication:

- Results section could be restructured to present the survey results in chronological order (e..g reorder lines 282-288).

- Table 2, interpretation would be made easier if all categories where grouped Jun 2020 - Jan 2021; and Jan 2021 - Apr 2022.

- Lines 84-85, Sao Paulo was perhaps the most important site of entry of SARS-CoV-2 in the country in the early phase of the pandemic as the authors mention. This has been best shown in Candido et al. JTM 2020 and Candido et al. Science 2021.

- Lines 367-369. Include 95% CIs.

- Lines 71. Compare and discuss results obtained from seroprevalence studies from Sao Paulo, including Buss et al. Science 2021 and Prete et al. eLife 2022.

Reviewers' comments:

Reviewer's Responses to Questions

**Comments to the Author**

1. Is the manuscript technically sound, and do the data support the conclusions?

Reviewer #1: Yes

2. Has the statistical analysis been performed appropriately and rigorously? 

Reviewer #1: Yes

3. Have the authors made all data underlying the findings in their manuscript fully available?

Reviewer #1: Yes

4. Is the manuscript presented in an intelligible fashion and written in standard English?

Reviewer #1: Yes

5. Review Comments to the Author

Reviewer #1: This study presents an important contribution to the literature. Its major strength is to look at SARS-CoV-2 antibodies in a representative sample of the population, and to estimate underreporting using infection-case calculations.

My main concern with this study is the omission of several important studies that have looked at (pre-vaccination) SARS-CoV-2 antibodies in Sao Paulo. Namely, the authors need to compare their findings to the studies from Buss et al Science 2021 and Prete et al. eLife 2022, which looked at SARS-CoV-2 antibodies in (more cost effective) blood donor populations in Sao Paulo city using Abbot assays. Another important limitation is the lack of adjustments regarding sensitivity and specificity and seroreversion (if any) for the Roche assay.

Other minor edits are also required before publication:

- Results section could be restructured to present the survey results in chronological order (e..g reorder lines 282-288).

- Table 2, interpretation would be made easier if all categories where grouped Jun 2020 - Jan 2021; and Jan 2021 - Apr 2022.

- Lines 84-85, Sao Paulo was perhaps the most important site of entry of SARS-CoV-2 in the country in the early phase of the pandemic as the authors mention. This has been best shown in Candido et al. JTM 2020 and Candido et al. Science 2021.

- Lines 367-369. Include 95% CIs.

- Lines 71. Compare and discuss results obtained from seroprevalence studies from Sao Paulo, including Buss et al. Science 2021 and Prete et al. eLife 2022.

6. PLOS authors have the option to publish the peer review history of their article (what does this mean?). If published, this will include your full peer review and any attached files.

Reviewer #1: No

---

## [Author Response · Author response to Decision Letter 0]

26 Oct 2023

We thank both the academic editor and the reviewer for their valuable comments/suggestions. We have revised the entire text according to those comments as presented below. 

Journal Requirements:

Response: We have adjusted our submitting materials to meet the PLOS ONE style templates.

Response: We have included the updated Competing Interests statement in our cover letter.

Response: We have included the information requested by the academic editor in the text (lines 174-178). Please refer to line 191 of the revised manuscript where we stated that “After participants signed the written informed consent, face-to-face interviews were conducted….”.

Response: Our dataset can be accessed at the University of São Paulo repository: https://sites.usp.br/epi-di/pagina-1/pagina-1-1/

 

Editor’s and reviewer’s comments:

5. The authors should consider several important studies that have looked at (pre-vaccination) SARS-CoV-2 antibodies in Sao Paulo. Namely, the authors need to compare their findings to the studies from Buss et al Science 2021 and Prete et al. eLife 2022, which looked at SARS-CoV-2 antibodies in (more cost effective) blood donor populations in Sao Paulo city using Abbot assays. 

Response: We have included both references in the text (lines 382 and 390, references number 26 and 27).

6. Another important limitation is the lack of adjustments regarding sensitivity and specificity and seroreversion (if any) for the Roche assay.

Response: We have added in the text (lines 214-216) the 95%CI for sensitivity (99.5% [95% CI: 97.0-100.0]) and specificity (99.8% [95% CI: 99.7-99.9]) according to the development and validation of the Elecsys Anti-SARS-CoV-2 Immunoassay (Muench et al, 2020). Considering these parameters, the adjustment in prevalence would not produce statistically significant difference in results. In addition, Roche Elecsys is less prone to antibody waning (Muecksch 2021). Furthermore, the seroprevalence trend was not affected since we applied the same assays throughout study period. The study design consisted of seven independent cross-sectional population-based surveys that does not allow to measure antibody waning. Nevertheless, we considered these issues in the discussion section (lines 481-484). 

Minor edits:

7. Results section could be restructured to present the survey results in chronological order (e..g reorder lines 282-288).

Response: We have made the changes to follow the chronological order.

8. Table 2, interpretation would be made easier if all categories were grouped Jun 2020 - Jan 2021; and Jan 2021 - Apr 2022. 

Response: The time series analysis used in the present study presents parameters and statistical results of the JoinPoint models for each variable. These models identified the number of turning points and the period (trend segments) for each variable. The results of these analyses were able to show distinct growth trends for different categories. In fact, the JointPoint analysis allows us to show changes in the seroprevalence trend. Grouping or fixing the periods is not suitable for time series analysis. For example, the growth model showed no turning point and only one period (straight line for the entire period Jun 2020 - Apr 2022) for the high-income strata. In contrast, the model identified one turning point for the low-income strata and two trend periods (June 2020-October 2020; October 2020-April 2022). In order to make the interpretation easier, we modified the statistical analysis (lines 246-258) and the included a footnote for period in Table 2 (lines 343-344).

9. Lines 84-85, Sao Paulo was perhaps the most important site of entry of SARS-CoV-2 in the country in the early phase of the pandemic as the authors mention. This has been best shown in Candido et al. JTM 2020 and Candido et al. Science 2021.

Response: We have included the two studies by Candido et al in the Introduction section (references number 7 and 8, line 85).

10. Lines 367-369. Include 95% CIs.

Response: We have included the 95% CI in line 369.

11. Lines 71. Compare and discuss results obtained from seroprevalence studies from Sao Paulo, including Buss et al. Science 2021 and Prete et al. eLife 2022.

Response: We have included a paragraph (lines 380-390, references number 26 and 27) in the Discussion section as suggested.

---

## [Decision Letter · Decision Letter 1]

3 Jun 2024

PONE-D-23-14879R1Seroprevalence trends of anti-SARS-CoV-2 antibodies in the adult population of the São Paulo Municipality, Brazil: Results from seven serosurveys from June 2020 to April 2022. The SoroEpi MSP StudyPLOS ONE

Dear Dr. TESS,

Thank you for submitting your manuscript to PLOS ONE. After careful consideration, we feel that it has merit but does not fully meet PLOS ONE’s publication criteria as it currently stands. Therefore, we invite you to submit a revised version of the manuscript that addresses the points raised during the review process.

The authors have worked hard on this reviewed manuscript, but some points need to be improved in order to be published. 

We look forward to receiving your revised manuscript.

Kind regards,

Marília Jesus Batista de Brito Mota, Post-doc

Academic Editor

PLOS ONE

Additional Editor Comments:

The authors have worked on this manuscript reviewed. The main limitation of the discussion is the comparison of the results of this study with the results of seroprevalence studies on SARS-CoV-2 among blood donors. These are different categories with distinct methodological approaches. To dispel doubts regarding potential "bias" and "confounding" in the study design, it is necessary to explain the SARS-CoV-2 prevalence values that inversely correlate with the monitoring period. All other comments are in the text of the paper.

Reviewers' comments:

Reviewer's Responses to Questions

**Comments to the Author**

1. If the authors have adequately addressed your comments raised in a previous round of review and you feel that this manuscript is now acceptable for publication, you may indicate that here to bypass the “Comments to the Author” section, enter your conflict of interest statement in the “Confidential to Editor” section, and submit your "Accept" recommendation.

Reviewer #2: (No Response)

Reviewer #3: All comments have been addressed

2. Is the manuscript technically sound, and do the data support the conclusions?

Reviewer #2: Yes

Reviewer #3: Yes

3. Has the statistical analysis been performed appropriately and rigorously? 

Reviewer #2: Yes

Reviewer #3: Yes

4. Have the authors made all data underlying the findings in their manuscript fully available?

Reviewer #2: Yes

Reviewer #3: Yes

5. Is the manuscript presented in an intelligible fashion and written in standard English?

Reviewer #2: Yes

Reviewer #3: Yes

6. Review Comments to the Author

Reviewer #2: The main limitation of the discussion is the comparison of the results of this study with the results of seroprevalence studies on SARS-CoV-2 among blood donors. These are different categories with distinct methodological approaches. To dispel doubts regarding potential "bias" and "confounding" in the study design, it is necessary to explain the SARS-CoV-2 prevalence values that inversely correlate with the monitoring period. All other comments are in the text of the paper.

Reviewer #3: I extend my gratitude for the efforts put in my the authors, now the manuscript is in better shape, the clarity of information presented has been ensured.

7. PLOS authors have the option to publish the peer review history of their article (what does this mean?). If published, this will include your full peer review and any attached files.

Reviewer #2: No

Reviewer #3: **Yes: **Dr. Suresh Yadav

---

## [Author Response · Author response to Decision Letter 1]

13 Jul 2024

Response to the reviewers

Manuscript Number PONE-D-23-14879R2

We thank the reviewers for their comments/suggestions. We have numbered the comments by Reviewer #2 following the order they were included in the body of the R1 manuscript. All comments were answered as presented below. We revised the entire text according to these comments. 

Note to Reviewer #2: The authors are grateful for the time and effort you dedicated to reviewing our manuscript. Your detailed feedback has been invaluable in improving the quality and clarity of the paper. We noticed that you reviewed the manuscript by the end of December 2023, according to the dates recorded in all boxes containing your comments. We only got your comments on June 3, 2024. Unfortunately, the submission process and handle of our manuscript by the journal was troublesome and slow. 

ABSTRACT

Note: We made small alterations in the abstract text to keep it within 300 words.

Comment 1: Line 50_R1: Add "in the pre-vaccination period".

Response: We have added the term as requested (Line 48_R2).

Comment 2: Line 55_R1: In comparison to whom? In comparison to what?

Response: We have rewritten part of the results section according to the reviewer´s comment (Lines 53-55_R2).

INTRODUCTION

Comment 3: Line 85_R1: It is not São Paulo route of viral spread. Please replace with the population of this city.

Response: We have rewritten the sentence in accordance with the reviewer’s request (Line 81_R2).

Comment 4: Line 111_R1: Specify the time period (before January 17, 2021.)

Response: We have specified the period as requested (Line 108_R2).

METHODS

Comment 5: Line 121_R1: Do you mean "9-20 September 2021" instead of "9-10 September 2021"?

Response: We have corrected the typo (Line 116_R2).

Comment 6: Line 123_R1: Why/How were these specific seven snapshot periods chosen?

Response: The dates of the surveys were established according to the progress of the epidemic in São Paulo and to include the pre and post vaccination periods (Lines 116-118_R2).

Comment 7: Line 141_R1: Add to Fig 2. data regarding the dominant variants of SARS-CoV-2 in accordance with the periods of their occurrence. As an example, I provide the reference below: Banjac J, Vuković V, Pustahija T, Bohucki N, Kovačević Berić D, Medić S, Petrović V, Ristić M. Epidemiological Characteristics of COVID-19 During Seven Consecutive Epidemiological Waves (2020-2022) in the North Bačka District, Serbia. Viruses. 2023;15, 2221.

Response: We have changed the title of Fig. 2 to “Timeline of COVID-19 cases and deaths, SARS-CoV-2 variants according to serosurveys, São Paulo, Brazil, 2020-2022” (Lines 139 and 140_R2).

We have included the predominant circulating variants in the footnote of Fig 2 as requested (Lines 141-143_R2). 

Comment 8: Line 162_R1: If the census was conducted after 2010 but before the completion of the seventh study snapshot, it may compromise the results of this research. When was the last population census in Brazil, or specifically in São Paulo?

Response: Brazil's last population census was conducted in 2022 instead of 2020, and data were available in 2023. This delay was due to the COVID-19 pandemic and political constrains. The sizes of the census tracts were updated at each field work and weights were introduced to deal with this issue.

Comment 9: Line 178_R1: How is it possible that the study commenced six months prior to obtaining ethical committee approval for its implementation?

Response: We have made it clearer in the text. The original protocol was approved on May 15, 2020. January 14, 2021, was the date when the protocol’s amendment was approved. (Lines 179-181_R2).

Comment 10: Line 201_R1: Have all four COVID vaccines been available since January 17, 2021 in São Paulo? Were all four vaccines available throughout the entire vaccination period in São Paulo?

Response: We have rewritten this part of the text to make it clearer. (Lines 204-210_R2).

In São Paulo State, four types of vaccines were used based on their availability and immunization policies. The first vaccines introduced in the early phase of the Brazilian COVID-19 campaign were Coronavac (Sinovac), an inactivated whole-virus vaccine, and Covishield (AstraZeneca). These were followed by Ad26.COV2.S (Janssen) and BNT162b2 (Pfizer-BioNTech) vaccines. However, not all vaccines were available throughout the vaccination campaign due to shortages and delays. We included the reference: Li SL, et al BMJ Open 2024;14:e076354. doi:10.1136/ bmjopen-2023-076354.

Comment 11: Line 220_R1: Please specify the values sensitivity and specificity of the "Maglumi assay"

Response: We have specified the values of the sensitivity and the specificity of the IgM and IgG Maglumi assays (Lines 226-227_R2).

RESULTS

We have rewritten the first paragraph of the results to make the text clearer (Lines 277-280_R2). 

Comment 12: Line 274_R1: Add the percentage.

Response: The percentage was included in the text (Line 278_R2).

Comment 13: Line 275_R1: Add the percentage.

Response: The percentage was included in the text (Line 279_R2). 

Comment 14: Line 276_R1: Add the percentage.

Response: The percentage was included in the text (Line 279_R2).

Comment 15: Line 276-7_R1: Delete this sentence.

Response: The sentence was deleted.

Comment 16: Line 297_R1: According to the results in Table 1, this is not correct. Seroprevalence in the lower-income stratum was 16.5%, and in the high-income stratum was 13% during Survey 2.

Response: We have included the word “statistically” to indicate that in all surveys there was a significant difference, except for Survey 2 which showed a p-value of 0.20 in the statistical test (Line 298_R2).

Comment 17: Line 299_R1: Add (≥ 60 years).

Response: We have added the term in the sentence (Line 302_R2).

Comment 18: Line 299_R1: In the Table 1 is 6.0-16.2.

Response: We have corrected the typo (Line 303_R2).

Comment 19: Line 301_R1: Add (18-39 years).

Response: We have added the term in the sentence (Line 304_R2).

Comment 20: Line 303_R1: And, in comparison to the Asian or Indigenous group?

Response: We could not make statistical inferences because the precision of the estimates was too low. We have added a sentence in the text to explain why we have not used the estimates of the Asian or Indigenous group in our analyses: The Asian or Indigenous group did not provide reliable estimates due to its small sample size and large 95% confidence intervals (Lines 307-308_R2). 

Comment 21: Line 304_R1: Add “significantly”.

Response: We have rewritten this part of the text to make it clearer (Line 306_R2).

Comment 22: Line 304_R1: According to the results in Table 1, this is not correct. The population with 16 or more years of education had lower seroprevalence in comparison with others across all seven serosurveys.

Response: We corrected the text according to the reviewer’s comment (Lines 309_R2).

Comment 23: Line 305_R1: It is necessary to add information and interpretation results indicating that in the Asian or Indigenous group, seroprevalence values do not increase exponentially over time, between 1 and 7 survey (9.7%, 8.5%, 11.8%, 9.3%, 28.7%, 45.5%, and 80.4%).

Response: We understand that it would have been interesting to study the Asian or Indigenous group, but the numbers were too small to allow for comparisons and interpretation of the results. Please refer to the response to comment 20.

Comment 24: Line 312_R1: In S2 Table, for the age group 40-59 years in Survey 7, the total vaccinated count is 340, while in Table 1, for the same age group and Survey 7, it is 360. Please explain this. 

Response: We have checked our calculations and there was a typo in S2 Table – the correct is 288 and not 268, therefore the correct number is 360. We have corrected it in the S2 Table.

Comment 25: Line 315_R1 – Self-reported race/skin color: Why do the sums of respondents according to race/skin color differ from the Overall respondents across Surveys?

Response: This is because self-reported race/skin color was a variable with missing data. We have added a footnote in Table 1 with the number of missing information in each survey (Line 326_R2).

Comment 26: Line 338_R1. Why is there no representation of the Asian or Indigenous population on Fig 3d?

Response: In comments 20 and 23, we have explained why Asian or Indigenous population was not presented in the modeling analysis.

DISCUSSION

Comment 27: Line 364_R1: The main limitation of the discussion is the comparison of the results of this study with the results of seroprevalence studies on SARS-CoV-2 among blood donors. These are different categories with distinct methodological approaches.

Response: The second and third paragraphs of the discussion section have been rewritten to respond to the reviewer’s comment (Lines 380-409_R2).

Comment 28: Line 417_R1: It is necessary to explain the possible reasons for the seroprevalence of the Asian or Indigenous group, which did not exhibit exponential increases across Surveys 1 to 7= 9.7%, 8.5%, 11.8%, 9.3%, 28.7%, 45.5%, and 80.4%.

Response: Asian or Indigenous population were not included in this analysis because the prevalences could not be interpreted due to small sample size. Please refer to comments 20 and 23.

Comment 29: Line 432_R1: At the end (Survey 7), the seroprevalence values by age groups are as follows: 18-39 = 85.8%; 40-59 = 75.3%, and 60+ = 65.4%. Additionally, the percentage of unvaccinated individuals by age groups is: 18-39 = 2%; 40-59 = 2.6%, and 60+ = 0%. Are the lower seroprevalence rates of SARS-CoV-2 in the 60+ age group compared to younger categories a result of vaccination with different vaccines and/or less frequent infections with the SARS-CoV-2 virus in older individuals, or something else?

Response: We understand that the results mentioned in your comment refer to the last survey and are presented in Table 1 and S2 Table. These results were part of the descriptive analysis and not of the modeling analysis (Line 432_R1). In this regard, the interpretation of the increased growth of prevalence was based on time series analysis, specifically for the 60+ age group. We also showed the differences in the slopes among the three age groups during the study period (see Figure 3c). The high slope in prevalence growth occurred concurrently with the prioritization of COVID-19 vaccination in the older population. This last sentence is in the discussion section (Lines 439-441_R2).

Considering your comment, in the descriptive analysis, the lower prevalence of SARS-CoV-2 throughout the period suggests that older individuals were less exposed to the virus. Of note, the 60+ group was a priority group to start vaccination with CoronaVac as shown in the S2 Table. However, the serological marker applied in our survey does not allow us to distinguish between natural infection with SARS-CoV-2 and vaccine-induced immunity by different types of vaccines.

Comment 30: Line 469_R1: To dispel doubts regarding potential "bias" and "confounding" in the study design, it is necessary to explain the SARS-CoV-2 prevalence values that inversely correlate with the monitoring period.

Response: We have included two sentences in the limitations section of the discussion about the potential bias related to the low response rate in our study (Lines 482-484_R2).

Comment 31: Line 487_R1: Additionally, the study data do not indicate the duration that has passed since the moment of natural infection (primary or reinfection) and/or vaccination to the point of serum collection for testing antibodies against the SARS-CoV-2 virus.

Response: The time of natural infection or the vaccine intake in relation to the blood collection was out of the scope of our study. We have included one sentence in the limitation section of the discussion section to make it clearer (Lines 498-499_R2).

---

## [Editor Report · Decision Letter 2]

8 Aug 2024

Seroprevalence trends of anti-SARS-CoV-2 antibodies in the adult population of the São Paulo Municipality, Brazil: Results from seven serosurveys from June 2020 to April 2022. The SoroEpi MSP Study

PONE-D-23-14879R2

Dear Dr. TESS,

We’re pleased to inform you that your manuscript has been judged scientifically suitable for publication and will be formally accepted for publication once it meets all outstanding technical requirements.

Kind regards,

Marília Jesus Batista de Brito Mota, Post-doc

Academic Editor

PLOS ONE

Additional Editor Comments (optional):

Dear authors,

First of all, I would like to apologize for the delay in the editorial process. It was happened due to the difficulty in finding reviewers for this manuscript.

However, the manuscript has been evaluated by reliable reviewers and, after the changes made by the authors, they have accepted the manuscript.

In view of the clear response letter from the authors and the changes made to the manuscript, and the decision of the reviewers, my decision as editor is to accept the manuscript for publication.
---

## [Editor Report · Acceptance letter]

14 Aug 2024

PONE-D-23-14879R2 

PLOS ONE

Dear Dr. TESS, 

I'm pleased to inform you that your manuscript has been deemed suitable for publication in PLOS ONE. Congratulations! Your manuscript is now being handed over to our production team.

Kind regards, 

on behalf of

Professor Marília Jesus Batista de Brito Mota 

Academic Editor

PLOS ONE